# A Comparative Study of the Effect of Including Full-Fat *Tenebrio molitor* for Replacing Conventional Ingredients in Practical Diets for *Dicentrarchus labrax* Juveniles

**DOI:** 10.3390/ani15020131

**Published:** 2025-01-08

**Authors:** Sara Flores-Moreno, Francisco Javier Alarcón-López, Antonio J. Coronel-Domínguez, Eugenia Zuasti, Ismael Hachero-Cruzado

**Affiliations:** 1Instituto Andaluz de Investigación y Formación Agraria, Pesquera, Alimentaria y de la Producción Ecológica (IFAPA), Centro El Toruño, Junta de Andalucía, Camino Tiro Pichón s/n, 11500 El Puerto de Santa María, Cádiz, Spain; sarafloresmoreno97@gmail.com (S.F.-M.); mariae.zuasti@juntadeandalucia.es (E.Z.); 2LifeBioencapsulation S.L., Parque Científico PITA, 04131 El Alquián, Almería, Spain; falarcon@ual.es; 3Departamento de Biología y Geología, Ceimar-Universidad de Almería, 04120 La Cañada de San Urbano, Almería, Spain; 4Beetle Genius 3.0, 41500 Alcalá de Guadaira, Sevilla, Spain; ajcordom@gmail.com; 5“Crecimiento Azul”, Centro IFAPA el Toruño, Unidad Asociada al CSIC, 11500 El Puerto de Santa María, Cádiz, Spain

**Keywords:** *Dicentrarchus labrax*, fatty acid composition, fishmeal, sustainable protein sources, *Tenebrio molitor*

## Abstract

Mealworm meal is considered a potential alternative ingredient for replacing vegetable and marine-derived feedstuffs in aquafeeds due to its low carbon footprint, high digestibility, and protein and micronutrient content. The aim of this piece of research was to evaluate the effects of partial substitution of fish and plant meals and oils with insect meals on the growth performance and tissue lipid composition in juvenile seabass. For this purpose, four diets were prepared with different inclusions of full-fat insect meals: two with 5 and 10% substituting mainly fishmeal and two others with 10 and 20% substituting the plant ingredients. The experimental diets were tested against a control diet in a 49-day feeding trial. The results showed that replacing plant ingredients with insect meals improved the growth performance and the muscle and liver lipid profile.

## 1. Introduction

In recent years, the rapid expansion of aquaculture, as well as its competition with other animal production systems, has led to a high demand for the raw materials used for feed production, such as fishmeal and fish oil. The increase in per capita fish consumption coupled with the stagnation in the global production of these natural resources is leading to a long-term increase in their prices [1]. All this only encourages the search for other dietary ingredients that make aquaculture a sustainable activity. In recent years, the most widely used alternatives to fishmeal have been plant protein sources, mainly soybean, rapeseed, corn, and wheat gluten, which are economically more profitable alternative sources of protein. However, their inclusion in carnivorous fish diets is limited due to unbalanced amino acid profile, fatty acid deficiency in n-3 HUFA, and the presence of antinutritional compounds which may interfere with the fish’s digestive processes [2]. Insects are becoming popular as potential substitutes for animal and vegetable ingredients, especially owing to that some of them are components of the natural diet of several species of freshwater aquaculture fish [3]. Compared with other regular feed ingredients, insects are more sustainable as they require fewer resources for their mass production [4], significantly reducing carbon footprint and land use, and can be reared as efficient biotransformers by using by-products, thus converting low-cost organic waste into animal biomass, rich in protein (54% to 70%), lipids (8% to 35%), essential amino acids, minerals, and B-group vitamins, suitable for using in aquafeeds [5,6,7,8].

Among the different insect species, the larvae of the mealworm (*Tenebrio molitor*, TM) stand out, characterised by the high biological value of their protein, fatty acid profile, and digestibility [4]. TM is rich in fat (34.5%) and crude protein (53.2%) [5,9]. Despite its limited content in sulphur-containing amino acids (cystine + methionine), TM has an adequate amino acid profile [5,10], and lysine and leucine are present in abundance [9]. TM is a good source of micronutrients, such as magnesium, iron, copper, manganese, zinc, selenium, riboflavin, biotin, pantothenic acid, and folic acid [5,11]. However, it may be deficient in vitamins A, B1, B12, D3, and E, iodine, manganese, calcium, and sodium [12]. TM lipids contain considerable amounts of unsaturated fatty acids, including the monounsaturated oleic acid (OA, 18:1n-9), which are key substrates in different metabolic processes in animals [13]. Nevertheless, its principal disadvantage is the absence of n-3 long-chain polyunsaturated fatty acids (LC-PUFA), including eicosapentaenoic acid (EPA, 20:5n-3) and docosahexaenoic acid (DHA, 22:6n-3) [14]. The potential of TM as a partial or total replacement for fishmeal (FM) in aquafeeds has been the subject of extensive evaluation, with particular attention paid to its influence on fish growth and nutrient utilisation in a diverse range of species, including Nile tilapia (*Oreochromis niloticus*) [15], rainbow trout (*Oncorhynchus mykiss*) [16,17], seabass (*Dicentrarchus labrax*) [18,19], gilthead seabream (*Sparus aurata*) [20], Atlantic salmon (*Salmo salar*) [21], and red seabream (*Pargus major*) [22], among others. Recent studies, such as those by Mastoraki et al. [18], Basto et al. [23], Ennayer et al. [24], and Basto et al. [25], showed the feasibility of replacing up to 50% of FM with TM in European seabass without compromising fish growth, liver health, intermediary metabolism, or nutrient digestibility. Reyes et al. [26] observed no differences in the feed conversion ratio (FCR) of *D. labrax* fed a diet with 50% TM substitution in a 49-day feeding trial, and Mastoraki et al. [18] reported a slight increase in the FCR in *D. labrax* fed a diet with 30% of FM replaced by TM for 84 days. Basto et al. [25] found that a total substitution of FM by TM did not affect fish growth but increased EPA and DHA levels in muscles, as well as promoted a better “juicy fillet texture” compared to FM-fed fish. However, other authors reported opposite results; for instance, Gasco et al. [27,28] found that 50% TM dietary inclusion negatively impacted the final body weight, WG (weight gain), and SGR in European seabass. Regarding the effects of TM meal on the fatty acid profile, Gasco et al. [28] and Mastoraki et al. [18] reported that the use of TM decreased long-chain n-3 polyunsaturated fatty acids but increased oleic and linoleic acids, which suggested a negative impact on the lipidic profile in fish fillets, including the n-3/n-6 ratio and the thrombogenicity index.

The discrepancies among the above-mentioned studies may be due to the type of insect meal used in the aquafeed formulation, defatted or full-fat TM meal. Several of the referenced studies employed defatted TM, but it should be considered that the process used for its production contributes to the enhancement of the ingredient’s carbon footprint and does not prevent the denaturation of nutrients. Moreover, the diets used as controls in those studies did not mimic the ingredient composition of commercial aquafeeds used these days, owing to that most of them used high and low dietary levels of FM and plant protein sources, respectively, just the opposite of the formulation standards. This is the reason why the present study aimed to assess, on one hand, the effects of replacing FM by TM, and on the other hand, replacing plant ingredients by TM, in both cases using a control diet formulated with a high inclusion level of plant protein ingredients. That way, it would be possible to ascertain the effects of a full-fat TM dietary inclusion in function of the type of feed ingredient replaced.

Therefore, this study aimed to evaluate the effects of a partial substitution of FM or plant ingredients by a full-fat *Tenebrio molitor* meal on the growth performance and fatty acid profile of lipids in the tissues of European seabass (*Dicentrarchus labrax*) juveniles. For this purpose, five experimental diets were formulated: a control diet with high plant protein content (CT) and four additional diets, two including insect meal at 5% (TM5) and 10% (TM10) for replacing mainly FM and soybean oil and two others including insect meal at 10% (PI10) and 20% (PI20) for partially replacing plant protein ingredients and soybean oil.

## 2. Materials and Methods

### 2.1. Insect Meal and Experimental Diets

Insect meal from *T. molitor* (TM) was provided by Beetle Genius S.L. (Brussels, Belgium). Proximate compositions and fatty acid (FA) profiles are detailed in Table 1 and Table 2. Five isolipidic (19%) and isonitrogenous (44%) diets containing TM were formulated (Table 1). Experimental diets FM5 and FM10 included 5% and 10% of TM, respectively, which mainly replaced FM, but also soybean oil; while diets PI10 and PI20 contained 10% and 20% of TM meal, respectively, replacing plant ingredients (protein and soybean oil). The control diet (CT) contained no TM meal. To achieve diets with equal lipid content, all diets with the TM meal were adjusted by reducing the content of soybean oil.

The formulation and preparation of the experimental diets were carried out by the Experimental Diets Service of the CEIMAR–University of Almeria (Almeria, Spain) using standard methods for aquaculture feed preparation. Briefly, all the ingredients were mixed in a 120 L mixer and ground using a hammer mill (UPZ 100, Hosokawa-Alpine, Augsburg, Germany) until particles of 0.5 mm were obtained. Subsequently, the diets were extruded using a twin-screw extruder (Evolum 25, Clextral, Firminy, France) equipped with dies suitable for the production of 2 and 3 mm sinking pellets. The extruder barrel consisted of four sections with temperature profiles of 100 °C, 95 °C, 95 °C, and 90 °C, respectively, along the sections. After extrusion, the pellets were dried at 30 °C in a 12 m3 drying chamber with forced air circulation (Airfrio, Almeria) and then cooled to room temperature. Vacuum oil coating was performed on the following day in a Pegasus PG-10VC LAB vacuum coater (Dinnissen, The Netherlands). Then, feeds were kept in sealed plastic bags at −20 °C until use.

The compositions of the experimental diets, including the formulation and chemical compositions, are presented in Table 1 and Table 2 and analysed according to Section 2.5. All diets exhibited a comparable level of total saturated FA (Table 2), although diets that included TM as a substitute for plant ingredients (PI10 and PI20) showed a higher content of 14:0 and palmitic acid (PA, 16:0). The most significant distinctions between the diets were in their total monounsaturated FA (MUFA) and n-3 and n-6 PUFA contents. The control diet (CT) showed the lowest levels of MUFA, in particular OA (18:1n-9), and the highest levels of n-6 PUFA and n-3 PUFA, mainly LA (18:2n-6) and LNA (18:3n-3), respectively.

No notable differences were found in terms of lipid class composition.

### 2.2. Fish and Experimental Designs

The European seabass juvenile stock (n = 2000) was provided by CUPIBAR, an aquaculture company located in Chiclana, Cádiz, Spain. These fish were transferred to IFAPA El Toruño, located in El Puerto de Santa María, Cádiz, Spain, where they were acclimatized in two 5000 L tanks for one month before starting the growth study. During this acclimatization period, the fish were fed a commercial dry feed (INTRO PLUS MT, Biomar, Dueñas, Spain) by automatic belt feeders, and no mortalities or signs of disease were recorded. For fish monitoring, one week after starting the acclimatisation period, all fish were anaesthetised (using 2-phenoxyethanol at 150 ppm), and a passive integrated transponder (PIT) (Trovan, 1.4 × 8 mm; Fish-Tags©, Melton, UK) was implanted intraperitoneally. They were then left for a further 3 weeks to monitor the healing of the tag wounds.

To analyse the impact of the experimental diets on growth, a total of 1.500 fish were randomly selected and redistributed in fifteen tanks (with a total volume of 500 L and three replicates per dietary treatment), maintaining an initial density of 100 fish per tank. The average weight at the start of the trial was 21.1 ± 4.9 g. Throughout the growth period, oxygen, salinity, and temperature levels were monitored daily and maintained in ranges of 4 to 6 ppm, 36 to 40 ppt, and 20 to 24 °C, respectively. The diets were administered through automatic Mirafeed © (Innovaqua, Lebrija, Spain) feeders in 25 daily portions, distributed between 08:00 and 20:00. The quantity of feed provided was adjusted on a weekly basis in line with the anticipated total biomass, calculated based on previous sampling (equivalent to 1.5–1.8% of the total biomass). In addition, the remaining daily feed in the tanks was considered to make adjustments in the amount of feed supplied. Individual fish weights were recorded every two weeks from the beginning of the trial until 49 days. Prior to each sampling event, specimens fasted for one day and were anaesthetised before any manipulation (using 2-phenoxyethanol at 150 ppm). Weight capture and PIT tag reading were carried out automatically using the FISH Reader (Zeus, Trovan, Spain) automated system.

### 2.3. Fish Sampling

The weight of each individual was recorded throughout the 49-day experiment. Four samplings were conducted at different time points: the initial measurement (t0), day 15 (t1), day 30 (t2), and day 49 (t3). Prior to each sampling session, the specimens underwent a one-day fasting period and were anesthetized using MS-222 at a concentration of 200 ppm. The recording of weight and PIT Tags were automated through the utilization of a FISH Reader Weight (Zeus, Trovan, Spain).

During the final sampling, thirty fish (six per diet) were euthanized using an overdose of anesthesia (MS-222 at >500 mg L^−1^). Muscle and liver samples were swiftly frozen in liquid nitrogen and preserved at −80 °C until further analysis. Liver and perivisceral fat were weighed to calculate the hepatosomatic (HSI) and perivisceral fat indexes (VFI).

### 2.4. Growth Performance and Nutrient Utilization

Growth performance was assessed using the following formulae: specific growth rate (SGR, % d^−1^) = (Ln (Wf) − Ln (Wi)/∆days) × 100, where Wf and Wi were the final and initial fish weight. Nutrient utilization indexes were estimated as follows: feed conversion ratio (FCR) = total feed intake on dry basis (g)/weight gain (g). Fulton condition factor (K) = 100 × (body weight)/fork length^3^).

### 2.5. Biochemical Analysis

#### 2.5.1. Proximate Compositions of Diets and Tissues

The proximate compositions of the diets (Table 1), including moisture, ash, lipid, and protein, as well as those of the fish tissues, in terms of lipids, were analysed following conventional procedures. Briefly, moisture content was determined by drying the sample in an oven at 110 °C for 24 h, while ash levels were measured in a muffle furnace at 600 °C for 16 h. The crude protein was measured by determination of the nitrogen content (N × 6.25) in agreement with AOAC [29] procedures. The total lipids were extracted from experimental fish diets (Table 2) and tissues of the experimental fish and quantified according to Folch et al. [30].

#### 2.5.2. Total Lipids, Lipid Classes, and FA Analyses

To extract lipids, approximately 200 mg of ground feed or fish tissues were placed in a mixture of ice-cold chloroform and methanol (in a ratio of 2:1 and by volume) and homogenized using an Ultra-Turrax tissue disrupter (Fisher Scientific, Loughborough, UK). Subsequently, the nonlipid and lipid layers were separated by the addition of 0.88% (*w*/*v*) KCl. The upper aqueous layer was aspirated and discarded, while the lower organic layer was dried under an oxygen-free nitrogen atmosphere. The lipid content was then determined by gravimetry after an overnight drying period in a vacuum desiccator.

Lipid classes were separated using one-dimensional double-development high-performance thin-layer chromatography (HPTLC) on 20 × 10 cm plates, following the method described by Olsen and Henderson [31]. Initially, the plates were impregnated with chloroform:methanol and subsequently dried at 120 °C for 1 h. The lipid classes were visualised by immersing the plates in a 3% (*w*/*v*) copper acetate containing 8% (*v*/*v*) phosphoric acid, followed by charring at 160 °C for 20 min. Quantification was carried out by densitometry using a CAMAG-3 TLC scanner (Firmware Version 1.14.16; CAMAG, Muttenz, Switzerland) with visionCATS CAMAG HPTLC SOFTWARE version 3.1. To identify individual lipid classes, samples and authentic standards were analysed together on HPTLC plates, contrasting the Rf values.

To extract FA, total lipid extracts were subjected to acid-catalysed transmethylation at 50 °C for 16 h according to Christie [32]. Fatty acids methyl esters (FAME) were separated and quantified by using a Shimadzu GC 2010-Plus gas chromatograph equipped with a flame ionization detector (280 °C) and a fused silica capillary column SUPRAWAX-280 (15 m × 0.1 mm I.D.) (Teknokroma, San Cugat del Valles, Spain). Hydrogen was used as a carrier gas, and the initial oven temperature was 100 °C for 0.5 min, followed by an increase at a rate of 20 °C min^−1^ to a final temperature of 250 °C for 8 min. Individual FAME were identified and quantified using an external standard (Sigma-Aldrich).

### 2.6. Statistical Analysis

A two-way repeated measures ANOVA was performed to test the effect of the five diets and the four sampling points on fish weight and length. Tank was added to the model as a random factor. A one-way ANOVA was performed to evaluate the effect of diets on initial and final fish weight, on the specific growth rate (SGR), nutritional utilization parameters (FCR, HSI and VFI), and on muscle fillets and liver protein and lipid composition at the end of trial. Where ANOVA indicated a significant difference (*p* < 0.05) for a given factor, the source of the difference was identified using a Tukey test. All statistical analyses were carried out using SPSS statistics v22 software (IBM, Armonk, USA) and GraphPadPrism 8.0.2 software for Windows. Before running all parametric tests, the normality was confirmed with a Kolmogorov–Smirnov test (*p* > 0.05) and the homogeneity of variances with a Levene test (*p* > 0.05). The proportions were transformed by arcsine transformation before analysis.

## 3. Results

### 3.1. Growth Performance

No mortality was found in any of the dietary treatments during the entire feeding trial. The fish weight and total length were monitored for 49 days using a longitudinal approach (Figure 1). A repeated measures ANOVA analysis identified a statistically significant interaction of diet × time for body weight and total length, which indicates that the increase in these parameters was different among dietary treatments. Despite no initial differences observed, animals fed on the PI20 diet were larger and heavier than those fed on the FM10 diet at the end of the feeding period (Table 3 and Figure 1). The specific growth rate (SGR) statistically differed among dietary groups (Figure 2), with the higher values in fish fed diets that included the TM for replacing plant ingredients (PI10 and PI20) and the lower for specimens fed diets that replaced mainly FM by TM (FM5 and FM10), respect to the CT group (Figure 2). Furthermore, no differences were observed in the feed conversion ratios (FCRs) and hepatosomatic indexes among dietary treatments. However, an increased perivisceral fat index was noted in fish fed the FM10 diet in comparison to the FM5 and PI20 groups.

### 3.2. Lipid Profile

Table 4 and Table 5 summarize the total lipid contents, lipid classes, and fatty acid profiles in the liver and muscle of fish fed the experimental diets. Liver lipids differed significantly between fish fed the CT and the other dietary treatments. The highest values were found in the CT group, followed by fish fed FM10, FM5, and PI20, and finally the PI10-fed specimens (Table 4). In muscle, no statistically significant differences were observed among the different experimental groups (Table 5).

Regarding lipid classes, total phospholipids did not differ in liver (Table 4). However, there were significant differences in TAG and SE, with higher values found in fish fed diets with the TM substituting fishmeal and plant ingredients, respectively. FFA and CHO content differed between groups FM5 and FM10. In muscle, all diets including insect meal increased total phospholipids, DAG, and FFA, and reduced TAG and SE, compared to the CT diet (Table 5).

Regarding the liver and muscle fatty acid (FA) profiles, PA (16:0), SA (18:0), OA (18:1n-9), LA (18:2n-6), and DHA (22:6n-3) were the most abundant FAs in those tissues of European seabass juveniles (Table 4 and Table 5). In liver, saturated fatty acid (SFA) content was higher in fish fed with FM10 and PI10 compared to the rest of dietary groups. Both 14:0 (myristic acid, MA) and 16:0 (palmitic acid, PA) fatty acids increased in individuals fed with these diets, with the exception of 14:0, which also was higher in animals fed the PI20 diet. Oleic acid (OA) was marginally higher in the PI10 and PI20 groups, though no significant differences were found. CT and FM5 batches showed the higher LA and LNA content in liver. As for the main n-3PUFA, the FM10 group exhibited the highest levels of eicosapentaenoic acid (EPA, 20:5n-3) and the lowest levels of docosahexaenoic acid (DHA, 22:6n-3) in liver. A lower n-3 PUFA/n-6 PUFA ratio was found in the CT group, with an increase in a dose-dependent manner related to TM inclusion. In muscle, higher levels of saturated fatty acids (14:0 and 16:0) and OA and lower levels of LA (18:2n-6) and LNA (18:3n-3) were observed in fish fed TM-based diets (Table 5). However, these changes were only significant when the TM inclusion was 10% or higher, independent of the type of feed ingredient that TM substituted. In the case of EPA and DHA, no significant differences were found between specimens fed the CT- and TM-based diets. However, there were differences in the DHA levels in individuals of the FM5 and PI10 groups compared to those fed the PI20 diet, whose values were lower. The highest n-3 PUFA/n-6 PUFA ratios were observed in the PI10 and PI20 batches, although the FM5 group differed significantly from the CT.

## 4. Discussion

In recent years, insects have attracted increased attention as novel dietary ingredients in aquaculture feeds, mainly due to their high protein content, functional properties, and environmental sustainability when compared with plant-based ingredients and fishmeal production [33]. However, a drawback associated with the use of insect meal is its fatty acid composition. In the present study, we explored the impacts of partially replacing fishmeal and plant ingredients (plant meals and soybean oil) with full-fat TM meal on the growth and tissue lipid profile in European seabass (*Dicentrarchus labrax*) juveniles. The control diet used in the present study was highly similar to the commercial feeds for European seabass, with over 50% of the feed composed of plant-based ingredients and approximately 30% derived from marine animal ingredients. Therefore, the present trial provides more updated information in terms of feed composition, managing to improve fish growth with alternative ingredients using much lower levels of fishmeal and fish oil in the control diet than in other previously published studies [18,23,25].

The growth performance observed in the present study was comparable to that reported in previous studies [18,34]. The results showed that a partial replacement of fishmeal by a full-fat TM meal (up to 10% of TM inclusion) led to a significant reduction in fish growth. Other authors did not report differences in growth when the TM was used for replacing fishmeal [18,23,25]. Mastoraki et al. [18] replaced 30% of a fishmeal with a full-fat TM in a control diet formulated with a high level of fishmeal (65%) and fish oil (10%), while Basto et al. [23,25] totally replaced fishmeal with defatted TM using a control diet containing 54% of marine derivates (40% fishmeal and 14% fish oil) without a negative effect on the growth performance of fish. The high content of fishmeal in those diets may explain why the replacement of fishmeal with TM did not affect growth performance in the above-mentioned studies. This finding contrasts to the results of the present study, wherein 30% of fishmeal was substituted with full-fat TM using a control diet formulated with 31.3% of ingredients derived from marine animals, considering both the protein and lipidic components. Another factor that may contribute to these differences with the aforementioned studies is the use of defatted TM combined with a high level of fish oil (14%), whereas in the present study we employed full-fat TM meal combined with a reduced level of dietary fish oil (8%).

In contrast, the results obtained evidenced that partial replacement of plant ingredients by full-fat *T. molitor* (TM) in European seabass diets (up to 20% of TM dietary inclusion) promoted a positive effect on fish growth as evidenced by the highest final weight and SGR values. It would be interesting to investigate if the inclusion of higher dietary levels of full-fat TM meal might be plausible in aquafeeds formulated with a high content of plant protein in order to know how this ingredient impacts the fatty acid profile of fish tissues, particularly in n-3 PUFA, when the dietary fish oil is reduced to below 8%. Further studies are requested in this sense for ascertaining in this aspect.

In relation to the impact of experimental diets on tissue lipid profiles, in all the dietary groups it was found that the use of full-fat TM resulted in a reduction in total lipid content in the liver, though this effect was not observed in the muscle. The reduction in lipid content was more evident when TM replaced the plant feedstuffs (PI10 and PI20). This finding is consistent with the lower levels of neutral lipids, triglycerides, and sterol ester, in the muscle of fish fed TM-based diets, which suggests that dietary inclusion of full-fat TM reduces lipid accumulation in fish tissues. Jeong et al. [35] and Hachero et al. [8] reported decreased muscle total lipids in olive flounder (*Paralichthys olivaceus*) and Senegalese sole (*Solea senegalensis*) fed diets formulated with insect meal. These authors hypothesized that the reduction in lipid accumulation may be attributable to the reduction in fat digestibility and lipid absorption provoked by the chitin from the TM. However, other studies have reported no changes in lipid content in red seabream (*Pagellus bogaraveo*) [36] or rainbow trout (*Oncorhynchus mykiss*) [17], using reference diets rich in marine derivates (75% and 55% of marine ingredients, respectively).

The impact on lipid accumulation in tissues observed in the present study seems to be modulated based on the type of feed ingredient that TM substituted. Triglyceride levels in the liver remained unchanged in the PI10 and PI20 groups but increased when TM replaced FM, just the opposite to that observed in the muscle. Interestingly, increased perivisceral fat was also observed in fish fed the FM10 diet. In contrast to the present study, Hachero et al. [8] observed a similar response regardless of the type of ingredient replaced (marine or plant-based ingredients). This discrepancy may be attributed to the different proportion of dietary marine ingredients used in each study. Whereas in our study the percentage was 31.3%, the feeds used by Hachero et al. [8] contained 46% of dietary ingredients derived from marine origin.

In terms of fatty acid profiles, both the liver and muscle of fish fed the TM meal-based diets exhibited increased oleic acid (18:1n-9), which is characteristic of insect meal, while linoleic acid (18:2n-6) and linolenic acid (18:3n-3) decreased. With regards to n-3 PUFA, no differences among fish fed the CT and TM-based diets were found. Hachero et al. [8] found similar results in Senegalese sole fed complete diets based on *T. molitor*, evidencing the same response whatever the fish species considered. However, other studies reported increased oleic and linoleic acids in TM-fed fish together with a decreased n-3 PUFA content [28,37]. In the present study, the dietary lipid proportioned to the inclusion of full-fat TM was offset by a reduction in soybean oil, which yielded a reduced dietary intake of linoleic acid, preventing any impact on the dietary n-3 PUFA. Moreover, this result could be explained by the selective deposition of the dietary DHA, as this fatty acid is usually accumulated in fish tissues at higher concentrations than those present in the diets due to their low ability to synthetize LC-PUFA [38]. Finally, the n-3/n-6 ratio, used as an index of lipid quality, was higher in both tissues, liver and muscle, of fish fed diets formulated with the TM. There was a noticeable variation between the fish fed the higher substitution of plant ingredients by the insect meal (PI20) compared to the CT group. This finding seems to indicate that replacement of the plant-based ingredients in diets for juvenile European seabass with full-fat insect meal (up to 20%) has a beneficial impact on the muscle quality in terms of fatty acid composition.

## 5. Conclusions

The results obtained evidence that inclusion of up to 20% of full-fat insect meal (*Tenebrio molitor*) to replace plant feedstuffs in diets for European seabass (*Dicentrarchus labrax*) juveniles is feasible without adversely affecting fish survival and improves fish growth. The TM-based diets reduced neutral lipids in the fish tissues, this reduction being noticeable when insect meal replaces plant feedstuffs. Despite the absence of n-3 PUFA in the TM, replacing plant ingredients (especially soybean oil) with full-fat TM maintains n-3 PUFA and DHA levels and improves the n-3/n-6 ratio in tissues. This study confirms that full-fat TM is a suitable alternative ingredient for replacing fishmeal or plant ingredients, being more recommendable in European seabass juveniles for substituting plant feedstuffs by TM in diets containing high dietary levels of plant ingredients.

## Figures and Tables

**Figure 1 animals-15-00131-f001:**
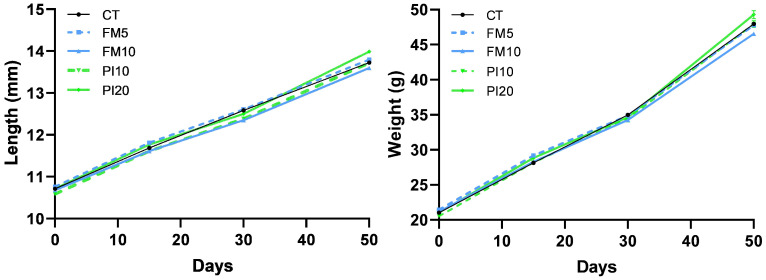
Evolution of body weight and length in European seabass juveniles fed with experimental diets. Dietary treatment codes are as follows: CT: control diet without TM meal; FM5: diet containing 5% TM meal, which mainly replaced fishmeal; FM10: diet containing 10% of TM meal, which mainly replaced fishmeal; PI10: diet containing 10% TM meal, replacing plant ingredients; PI20: diet containing 20% TM meal, replacing plant ingredients. Values are means of triplicate tanks.

**Figure 2 animals-15-00131-f002:**
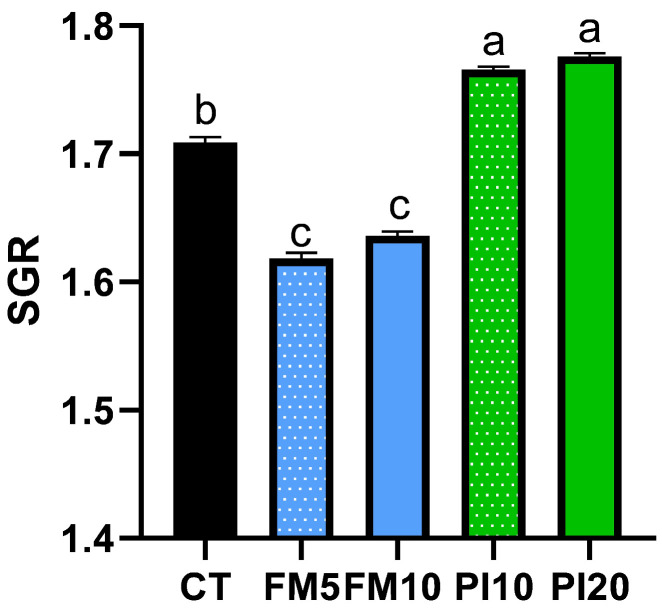
Specific growth rate of European seabass juveniles fed with the experimental diets. Dietary treatment codes are as follows: CT: control diet; FM5: diet containing 5% TM meal, which mainly replaced fishmeal; FM10: diet containing 10% of TM meal, which mainly replaced fishmeal; PI10: diet containing 10% TM meal, replacing plant ingredients; PI20: diet containing 20% TM meal, replacing plant ingredients. Values are mean ± SD. Values with different letters indicate significant differences among dietary treatments (*p* < 0.05).

**Table 1 animals-15-00131-t001:** Ingredient composition of experimental diets used for feeding *Dicentrarchus labrax* juveniles in the present study.

Ingredients (% dw)	TM	CT	FM5	FM10	PI10	PI20
Fishmeal LT94 ^1^		20.0	16.5	13.1	20.0	20.0
Squid meal ^2^		1.0	1.0	1.0	1.0	1.0
CPSP90 ^3^		1.0	1.0	1.0	1.0	1.0
Krill meal ^4^		1.0	1.0	1.0	1.0	1.0
Wheat gluten ^5^		11.0	11.0	11.0	9.3	7.4
Soybean protein concentrate ^6^		23.0	23.0	23.0	20.2	18.2
Pea protein concentrate ^7^		6.0	6.0	6.0	4.3	2.3
Full-fat *Tenebrio molitor* ^8^			5.0	10.0	10.0	20.0
Fish oil ^9^		8.3	8.3	8.3	8.3	8.3
Soybean oil ^10^		7.0	5.7	4.3	3.5	0.1
Soybean lecithin ^11^		1.0	1.0	1.0	1.0	1.0
Wheat flour ^12^		12.5	12.3	12.1	12.2	11.5
Choline chloride ^13^		0.5	0.5	0.5	0.5	0.5
Betaine ^14^		0.5	0.5	0.5	0.5	0.5
Vitamin and Mineral premix^15^		2.0	2.0	2.0	2.0	2.0
Vitamin C ^16^		0.1	0.1	0.1	0.1	0.1
Lysine ^17^		1.5	1.5	1.5	1.5	1.5
Methionine ^18^		0.6	0.6	0.6	0.6	0.6
Monoammonium phosphate ^19^		1.0	1.0	1.0	1.0	1.0
Guar gum ^20^		2.0	2.0	2.0	2.0	2.0
Crude protein (% dw)	43.77	46.7	47.0	47.2	47.3	47.8
Crude lipid (% dw)	35.05	21.8	21.1	21.5	21.0	21.8
Ash (% dw)	2.92	5.52	5.98	6.16	5.80	5.16
Moisture (% dw)	7.78	8.35	9.07	9.49	8.21	7.84

Dietary codes: CT: control diet without TM meal; FM5: diet containing 5% TM meal, which mainly replaced fishmeal; FM10: diet containing 10% of TM meal, which mainly replaced fishmeal; PI10: diet containing 10% TM meal, replacing mostly plant proteins; PI20: diet containing 20% TM meal, replacing mostly plant proteins. TM: *Tenebrio molitor*, ^1^ composed of 12.3% crude lipid and 69.4% crude protein (Norsildemel, Bergen, Norway), ^2, 3, 4^ was purchased from Bacarel (UK). CPSP90 is enzymatically predigested fishmeal, ^5^ composed of 78% crude protein (Lorca Nutrición Animal SA, Murcia, Spain). Total P: 0.718 g and 100 g. Total phytate P: 0.01 g and 100 g. ^6^ Soycomil: composed of 60% crude protein and 1.5% crude lipid (ADM, Poland). Total P: 0.66 g and 100 g. Total phytate P: 0.42 g and 100 g. ^7^ Pea protein concentrate: composed of 85% crude protein and 1.5% crude lipid (Emilio Peña SA, Spain). ^8^ Beetle Genius S.L. (Brussels, Belgium). ^9^ AF117DHA (Afamsa, Spain). ^10^ Soybean oil (Aceites el Niño, Spain). ^11^ P700IP (Lecico, DE). ^12^ Local provider (Almería, Spain). ^13, 14, 17, 18, 19^ Lorca Nutrición Animal SA (Murcia, Spain). ^15^ Lifebioencapsulation S.L. (Almería, Spain). Vitamins (mg kg^−1^): vitamin A (retinyl acetate), 2,000,000 UI; vitamin D3 (DL-cholecalciferol), 200,000 UI; vitamin E (Lutavit E50), 10,000 mg; vitamin K3 (menadione sodium bisulfite), 2500 mg; vitamin B1 (thiamine hydrochloride), 3000 mg; vitamin B2 (riboflavin), 3000 mg; calcium pantothenate, 10,000 mg; nicotinic acid, 20,000 mg; vitamin B6 (pyridoxine hydrochloride), 2000 mg; vitamin B9 (folic acid), 1500 mg; vitamin B12 (cyanocobalamin), 10 mg; vitamin H (biotin), 300 mg; inositol, 50,000 mg; betaine (Betafin S1), 50,000 mg. Minerals (mg kg^−1^): Co (cobalt carbonate), 65 mg; Cu (cupric sulfate), 900 mg; Fe (iron sulfate), 600 mg; I (potassium iodide), 50 mg; Mn (manganese oxide), 960 mg; Se (sodium selenite), 1 mg; Zn (zinc sulfate), 750 mg; Ca (calcium carbonate), 18.6% (186,000 mg); KCl, 2.41% (24,100 mg); NaCl, 4.0% (40,000 mg). ^16^ TECNOVIT, Spain. ^20^ EPSA, Spain.

**Table 2 animals-15-00131-t002:** Lipid classes (% TL) and fatty acid (% total FA) compositions of *T. molitor* (TM) meal and the experimental diets (CT, FM5, FM10, PI10, PI20).

			Diets			
	TM	CT	FM5	FM10	PI10	PI20
*Lipid classes (%TL)*						
Total phospholipids	5.9	8.24	9.9	9.6	9.2	8.7
CHO	7.4	10.8	12.1	12.0	11.0	12.5
FFA	34.6	9.9	13.0	12.6	13.1	12.5
TAG	43.4	49.4	41.3	48.6	47.4	44.3
SE	4.1	1.5	3.2	1.4	2.9	3.0
*Fatty acids % (TFA)*						
14:0	4.2	1.2	1.3	1.4	1.7	2.1
16:0, PA	18.0	16.0	16.1	16.5	17.4	18.1
18:0	2.5	4.9	4.6	4.7	4.6	4.3
Total saturated FA	24.7	23.6	23.6	24.0	25.2	25.9
16:1n-7	2.8	2.4	2.4	2.5	2.9	3.3
18:1n-9, OA	51.7	19.0	22.1	25.5	25.2	30.9
18:1n-7	0.1	2.2	2.0	2.0	2.3	2.0
20:1n-9	0.1	1.0	0.9	0.9	1.0	0.9
22:1n-11	nd	0.4	0.4	0.4	0.4	0.4
Total MUFA	55.9	25.8	28.9	32.5	33.1	39.0
18:2n-6, LA	18.0	29.9	27.3	23.8	21.7	15.8
20:4n-6, ARA	nd	0.8	0.8	0.9	0.8	1.0
22:5n-6	nd	0.6	0.6	0.6	0.6	0.6
Total n-6 PUFA	18.0	31.8	29.1	25.7	23.5	17.8
18:3n-3, LNA	0.3	3.4	2.9	2.2	2.1	1.2
18:4n-3	nd	0.4	0.4	0.3	0.4	0.4
20:5n-3, EPA	nd	3.4	3.2	3.1	3.4	3.3
22:6n-3, DHA	nd	9.0	9.7	9.5	10.0	9.6
Total n-3 PUFA	0.3	17.3	17.2	16.2	17.0	15.5
n-3PUFA/n-6PUFA	0.02	0.54	0.59	0.63	0.72	0.87
EPA/DHA	0.00	0.39	0.33	0.32	0.34	0.34
UFA/SFA	3.00	3.21	3.23	3.14	2.95	2.83

TM: *Tenebrio molitor*; CHO: cholesterol; FFA: free fatty acids; TAG: triglycerides; SE: sterols; TFA: total fatty acids; PA: palmitic acid; OA: oleic acid; LA: linoleic acid; FA: fatty acids; ARA: arachidonic acid; LNA: A-linolenic acid; EPA: eicosapentaenoic acid; DHA: docosahexaenoic acid; MUFA: monounsaturated fatty acids; PUFA: polyunsaturated fatty acids; UFA: unsaturated fatty acids; SFA: saturated fatty acids; nd: not detected. Dietary codes: CT: control diet without TM meal; FM5: diet containing 5% TM meal, which mainly replaced fishmeal; FM10: diet containing 10% of TM meal, which mainly replaced fishmeal; PI10: diet containing 10% TM meal, replacing mostly plant proteins; PI20: diet containing 20% TM meal, replacing mostly plant proteins.

**Table 3 animals-15-00131-t003:** Somatic indexes and nutrient utilization parameters of juvenile European seabass fed with the experimental diets during the 49-day feeding trial. Values are presented as mean ± SD. Values in the same row with different superscript letters indicate significant differences among dietary treatments (*p* < 0.05).

	CT	FM5	FM10	PI10	PI20
Initial body weight (g)	21.16 ± 4.89	21.61 ± 4.64	21.23 ± 4.90	20.5 ± 4.92	21.1 ± 4.85
Final body weight (g)	48.45 ± 8.39 ^ab^	47.89 ± 8.61 ^ab^	46.61 ± 8.35 ^b^	48.01 ± 8.12 ^ab^	49.57 ± 7.85 ^a^
Initial body length (cm)	10.69 ± 0.74	10.76 ± 0.71	10.70 ± 0.79	10.59 ± 0.76	10.72 ± 0.73
Final body length (cm)	13.71 ± 0.85 ^ab^	13.80 ± 0.88 ^ab^	13.62 ± 0.88 ^b^	13.72 ± 0.86 ^ab^	13.99 ± 0.82 ^a^
Fulton’s condition factor	1.87 ± 0.13	1.81 ± 0.14	1.83 ± 0.13	1.85 ± 0.14	1.80 ± 0.11
Feed conversion ratio (FCR)	1.11± 0.21	1.08 ± 0.11	1.11 ± 0.13	1.06 ± 0.19	1.02 ± 0.11
Hepatosomatic index (HSI, %)	1.39 ± 0.26	1.20 ± 0.40	1.21 ± 0.23	1.12 ± 0.24	1.23 ± 0.28
Perivisceral fat index (VFI, %)	3.90 ± 0.61 ^ab^	3.30 ± 1.24 ^b^	4.85 ± 0.74 ^a^	3.71 ± 0.94 ^ab^	3.41 ± 0.75 ^b^

CT: control diet without TM meal; FM5: diet containing 5% TM meal, which mainly replaced fishmeal; FM10: diet containing 10% of TM meal, which mainly replaced fishmeal; PI10: diet containing 10% TM meal, replacing plant ingredients; PI20: diet containing 20% TM meal, replacing plant ingredients.

**Table 4 animals-15-00131-t004:** Total lipid contents: lipid classes and fatty acid compositions (%TFA) in liver of European seabass fed the experimental diets (mean ± SD and n = 6). Values are presented as mean ± SD. Values in the same row with different letters indicate significant differences among dietary treatments (*p* < 0.05).

	Dietary Treatments				
	CT	FM5	FM10	PI10	PI20
Total lipids % (dw)	54.94 ± 0.36 a	45.28 ± 0.46 c	52.03 ± 0.47 b	36.32 ± 0.35 d	47.07 ± 0.57 c
Lipid class (%TL)					
Total phospholipids	27.74 ± 1.47	25.79 ± 2.01	25.53 ± 1.13	27.59 ± 0.92	25.87 ± 2.48
FFA	9.44 ± 0.18 ab	10.05 ± 0.92 a	8.79 ± 0.51 b	9.18 ± 0.85 ab	9.81 ± 0.18 ab
DAG	5.24 ± 0.32	4.77 ± 0.77	4.78 ± 0.56	4.90 ± 0.51	5.04 ± 0.54
CHO	7.00 ± 0.60 ab	7.40 ± 0.67 a	6.28 ± 0.47 b	6.81 ± 0.57 ab	6.62 ± 0.39 ab
TAG	42.43 ± 1.84 b	46.86 ± 2.03 a	46.61 ± 1.12 a	43.34 ± 0.80 b	44.65 ± 1.46 ab
SE	1.61 ± 0.41 b	1.87 ± 0.11 b	3.30 ± 0.66 ab	3.47 ± 0.65 a	3.98 ± 0.14 a
Fatty acids % (TFA)					
14:0	1.18 ± 0.12 c	1.26 ± 0.16 bc	1.32 ± 0.20 ab	1.46 ± 0.18 ab	1.55 ± 0.07 a
16:0. PA	18.62 ± 1.44 b	18.44 ± 1.31 b	20.80 ± 2.05 a	20.15 ± 1.61 a	18.31 ± 1.89 b
18:0	6.44 ± 0.53 a	5.98 ± 0.65 ab	6.99 ± 0.67 a	6.06 ± 1.11 a	5.02 ± 0.40 b
Total saturated FA	29.21 ± 1.40 b	28.63 ± 1.14 b	32.39 ± 2.05 a	30.46 ± 2.39 a	27.87 ± 1.92 b
16:1n-9	0.55 ± 0.06 b	0.62 ± 0.06 b	0.74 ± 0.14 b	0.80 ± 0.08 a	0.90 ± 0.07 a
16:1n-7	3.01 ± 0.29	3.15 ± 0.33	3.46 ± 0.59	3.5 ± 0.23	3.70 ± 0.28
18:1n-9. OA	32.49 ± 3.42	33.18 ± 5.71	35.43 ± 3.13	37.46 ± 4.20	38.21 ± 1.96
18:1n-7	0.07 ± 0.03	0.07 ± 0.03	0.04 ± 0.04	0.07 ± 0.00	0.06 ± 0.03
22:1n-11	0.08 ± 0.04 a	0.09 ± 0.05 a	0.05 ± 0.05 b	0.09 ± 0.01a	0.11 ± 0.02 a
Total monounsaturated FA	36.89 ± 3.84 b	37.76 ± 6.87 b	40.52 ± 3.57 a	42.80 ± 5.65 a	43.77 ± 3.98 a
18:2n-6. LA	15.47 ± 2.12 a	15.26 ± 3.04 a	10.17 ± 1.48 b	11.16 ± 2.66 b	10.18 ± 1.78 b
18:3n-6	0.77 ± 0.19 b	0.73 ± 0.24 b	1.33 ± 0.66 a	0.85 ± 0.08 b	0.65 ± 0.11 b
20:2n-6	0.51 ± 0.06 a	0.49 ± 0.09 a	0.33 ± 0.03 b	0.40 ± 0.06 b	0.38 ± 0.06 b
20:3n-6	0.97 ± 0.20 b	0.84 ± 0.30 b	1.40 ± 0.89 a	0.78 ± 0.10 b	0.88 ± 0.12 b
20:4n6. ARA	0.85 ± 0.03 b	0.80 ± 0.02 b	0.93 ± 0.05 a	0.80 ± 0.00 b	0.97 ± 0.07 a
22:4n-6	0.33 ± 0.05	0.47 ± 0.07	0.32 ± 0.05	0.33 ± 0.08	0.41 ± 0.06
22:5n-6	1.11 ± 0.24	1.01 ± 0.27	1.29 ± 0.66	0.85 ± 0.16	1.06 ± 0.15
Total n-6 polyunsaturated FA	19.15 ± 2.76 a	18.65 ± 3.82 a	14.64 ± 2.19 b	14.75 ± 3.07 b	13.68 ± 2.10 c
18:3n-3. LNA	1.39 ± 0.15 a	1.33 ± 0.30 a	0.73 ± 0.17 b	0.85 ± 0.26 b	0.65 ± 0.15 b
20:3n-3	0.10 ± 0.01 ab	0.11 ± 0.03 a	0.04 ± 0.04 b	0.09 ± 0.03 ab	0.12 ± 0.03 a
20:4n-3	1.91 ± 0.12 ab	1.94 ± 0.37 ab	1.44 ± 0.29 b	1.73 ± 0.32 ab	2.15 ± 0.32 a
20:5n-3. EPA	0.97 ± 0.30 b	0.75 ± 0.34 b	1.55 ± 1.10 a	0.65 ± 0.11 c	0.74 ± 0.12 bc
22:5n-3	0.11 ± 0.01	0.08 ± 0.01	0.06 ± 0.02	0.10 ± 0.01	0.09 ± 0.01
22:6n-3. DHA	6.19 ± 1.29 ab	6.59 ± 1.69 a	4.98 ± 1.28 b	5.44 ± 1.11 ab	6.94 ± 1.11 a
Total n-3 polyunsaturated FA	10.17 ± 1.82	10.27 ± 2.51	9.33 ± 1.77	9.34 ± 1.94	10.92 ± 1.58
n-3PUFA/n-6PUFA	0.60 ± 0.04 c	0.62 ± 0.04 bc	0.69 ± 0.02 b	0.66 ± 0.02 b	0.85 ± 0.02 a

DAG: diacylglycerol; CHO: cholesterol; FFA: free fatty acids; TAG: triglycerides; SE: sterols; TFA: total fatty acids; PA: palmitic acid; OA: oleic acid; LA: linoleic acid; FA: fatty acids; ARA: arachidonic acid; LNA: A-linolenic acid; EPA: eicosapentaenoic acid; DHA: docosahexaenoic acid; PUFA: polyunsaturated fatty acids. Dietary codes: CT: control diet without TM meal; FM5: diet containing 5% TM meal, which mainly replaced fishmeal; FM10: diet containing 10% of TM meal, which mainly replaced fishmeal; PI10: diet containing 10% TM meal, replacing mostly plant proteins; PI20: diet containing 20% TM meal, replacing mostly plant proteins.

**Table 5 animals-15-00131-t005:** Total lipid contents: lipid classes and fatty acid compositions (%TFA) in the muscle of European seabass fed the experimental diets (mean ± SD. n = 6). Values are presented as mean ± SD. Values in the same row with different letters indicate significant differences among dietary treatments (*p* < 0.05).

	Dietary Treatment				
	CT	FM5	FM10	PI10	PI20
Total lipids %	13.02 ± 0.50	12.08 ± 2.34	13 ± 2.44	12.76 ± 1.80	12.97 ± 1.67
Lipid classes (%TL)					
Total phospholipids	24.78 ± 1.71 b	32.46 ± 0.36 a	32.80 ± 0.99 a	31.98 ± 4.34 a	29.42 ± 3.89 ab
FFA	11.12 ± 1.85 b	12.58 ± 0.73 ab	12.78 ± 1.19 ab	14.06 ± 0.46 a	14.41 ± 0.32 a
DAG	3.39 ± 0.22 c	3.64 ± 0.29 ab	3.57 ± 0.25 b	3.71 ± 0.32 a	3.83 ± 0.32 a
CHO	10.03 ± 0.05	9.78 ± 0.08	9.85 ± 0.01	9.88 ± 0.09	10.29 ± 0.12
TAG	43.04 ± 2.52 a	36.46 ± 3.85 b	36.53 ± 1.30 b	34.63 ± 2.66 b	35.88 ± 3.67 b
SE	3.96 ± 0.86 a	2.47 ± 1.01 b	1.26 ± 0.45 c	2.38 ± 0.37 b	0.71 ± 0.32 c
Fatty acids % (TFA)					
14:0	1.68 ± 0.06 b	1.65 ± 0.07 b	1.83 ± 0.09 b	1.93 ± 0.06 a	2.01 ± 0.26 a
16:0. PA	16.49 ± 0.41 b	16.71 ± 0.47 b	17.65 ± 0.54 ab	17.17 ± 0.66 ab	17.95 ± 1.17 a
18:0	4.33 ± 0.20	4.36 ± 0.21	4.24 ± 0.21	4.79 ± 0.49	4.19 ± 0.23
Total saturated FA	23.85 ± 0.42 b	24.16 ± 0.57 ab	24.97 ± 0.74 ab	24.41 ± 1.22 ab	25.43 ± 1.21 a
16:1n-7	3.70 ± 0.26	3.29 ± 0.17	3.61 ± 0.24	3.59 ± 0.29	3.81 ± 0.50
18:1n-9. OA	23.58 ± 1.26 b	24.54 ± 0.83 b	28.92 ± 1.23 a	27.39 ± 2.53 a	27.98 ± 3.06 a
18:1n-7	3.26 ± 0.38	3.16 ± 0.22	2.96 ± 0.58	3.59 ± 0.29	3.74 ± 0.52
22:1n-11	0.74 ± 0.11 a	0.71 ± 0.09 a	0.59 ± 0.05 b	0.69 ± 0.07 a	0.87 ± 0.08 a
Total monounsaturated FA	33.89 ± 1.85 b	34.27 ± 1.16 b	38.79 ± 1.73 a	37.92 ± 3.08 a	39.23 ± 3.42 a
18:2n-6. LA	21.74 ± 1.52 a	20.10 ± 1.02 a	17.13 ± 1.82 b	17.18 ± 1.07 b	15.29 ± 0.38 c
18:3n-6	0.20 ± 0.03 a	0.18 ± 0.02 ab	0.16 ± 0.02 b	0.17 ± 0.02 b	0.13 ± 0.00 c
20:2n-6	0.63 ± 0.05 a	0.58 ± 0.06 ab	0.53 ± 0.07 b	0.58 ± 0.04 ab	0.52 ± 0.04 b
20:3n-6	0.10 ± 0.01	0.11 ± 0.03	0.09 ± 0.00	0.11 ± 0.01	0.10 ± 0.01
20:4n6. ARA	0.71 ± 0.02	0.82 ± 0.09	0.76 ± 0.02	0.78 ± 0.18	0.79 ± 0.16
22:4n-6	0.47 ± 0.03	0.52 ± 0.03	0.48 ± 0.06	0.51 ± 0.08	0.49 ± 0.05
Total n-6 polyunsaturated FA	23.76 ± 1.39 a	22.32 ± 0.95 a	19.16 ± 1.69 b	19.33 ± 0.97 b	15.84 ± 0.45 c
18:3n-3. LNA	2.56 ± 0.14 a	2.26 ± 0.13 a	1.89 ± 0.35 b	1.88 ± 0.13 b	1.61 ± 0.17 c
20:3n-3	0.10 ± 0.01	0.11 ± 0.00	0.10 ± 0.00	0.10 ± 0.00	0.10 ± 0.00
20:4n-3	0.27 ± 0.01	0.26 ± 0.01	0.25 ± 0.01	0.25 ± 0.01	0.26 ± 0.02
20:5n-3. EPA	3.50 ± 0.22	3.35 ± 0.21	3.22 ± 0.25	3.27 ± 0.15	3.25 ± 0.10
22:5n-3	0.79 ± 0.02	0.81 ± 0.04	0.73 ± 0.05	0.69 ± 0.06	0.77 ± 0.03
22:6n-3. DHA	8.69 ± 0.70 ab	9.89 ± 0.95 a	8.74 ± 0.95 ab	9.50 ± 0.78 a	8.55 ± 0.68 b
Total n-3 polyunsaturated FA	16.14 ± 0.75	16.94 ± 1.08	15.13 ± 1.35	15.92 ± 1.99	14.93 ± 1.20
n-3PUFA/n-6PUFA	0.67 ± 0.04 c	0.76 ± 0.04 b	0.70 ± 0.08 c	0.82 ± 0.04 b	0.94 ± 0.05 a

DAG: diacylglycerol; CHO: cholesterol; FFA: free fatty acids; TAG: triglycerides; SE: sterols; TFA: total fatty acids; PA: palmitic acid; OA: oleic acid; LA: linoleic acid; FA: fatty acids; ARA: arachidonic acid; LNA: A-linolenic acid; EPA: eicosapentaenoic acid; DHA: docosahexaenoic acid; PUFA: polyunsaturated fatty acids. Dietary codes: CT: control diet without TM meal; FM5: diet containing 5% TM meal, which mainly replaced fishmeal; FM10: diet containing 10% of TM meal, which mainly replaced fishmeal; PI10: diet containing 10% TM meal, replacing mostly plant proteins; PI20: diet containing 20% TM meal, replacing mostly plant proteins.

## Data Availability

The original contributions presented in the study are included in the article, further inquiries can be directed to the corresponding author.

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
