# Peer review of "A Comparative Study of the Effect of Including Full-Fat Tenebrio molitor for Replacing Conventional Ingredients in Practical Diets for Dicentrarchus labrax Juveniles"

_animals, 2025, doi:10.3390/ani15020131_

Round 1

Reviewer 1 Report

Comments and Suggestions for Authors

L15-16 Which author is the corresponding author? Please provide his email address and phone number.

L30-34 TM diets or FM diets, which can lead to confusion. It is recommended that FM diets are used for the entire manuscript.

L35-36 Tissue quality. This study did not provide data related to tissue quality, only results for fatty acid composition, so it is better to conclude trends in specific fatty acids herein rather than using “tissue quality”.

L87 Increased or decreased EPA and DHA in muscle? The author of the above text explains that TM is deficient in EPA and DHA?

L226 feed conversion ratio (FCR) = total feed intake on dry basis (g) weight gain (g)-1. Please revise this equation. feed conversion ratio (FCR) = total feed intake on dry basis (g) × weight gain (g)-1 or feed conversion ratio (FCR) = total feed intake on dry basis (g) / weight gain (g)

L232 ... ash levels were calculated... It should be “measured or determined” not “calculated”.

L291 Table 3 The letters “ab” need to be superscripted.

L311 The letter “P” needs to be italicized.

L360 Currently, as insects have attracted increasing… Please delete “as”.

L374-376 Insects belong to animal proteins, which can be used as alternative ingredients to plant proteins to improve growth in carnivorous fish, and this result is expected. However, the price of insect protein is higher than that of plant protein, and even if the substitution is successful, it cannot be used in large quantities in commercial feed formulations. Therefore, the hypothesis of this study is debatable.

L405 The use of full-fat TM reduced total lipid content in the liver. The reasons for this result should be further explained in the following discussion.

L426 18:3n3 or 18:3n-3?

L428 T. molitor. It should be italicized.

L441 Two spaces at the beginning of the paragraph and the initial letter cannot be bold.

L444-445 This result is actually caused by oil from TM replacing soybean oil, not by insect proteins replacing plant proteins.

Reviewer 2 Report

Comments and Suggestions for Authors

In attachment you will find all the comments related with the manuscript.

Reviewer 3 Report

Comments and Suggestions for Authors

The paper concerns the utilization of insect, in this case Tenebrio molitor (TM), as a substitute ingredient of plant protein and fishmeal in Dicentrarchus labrax juveniles’ diets. In detail, four diets (plus a control diet) were tested, two replacing plant protein (10-20%) and two replacing fishmeal (5-10%). Authors analyzed different parameters, such as growth performance, liver and muscles lipid content and profile. Results demonstrate that TM inclusions in the diets improved tissue quality and growth performance when substitute plant protein, whereas showed a slightly reduction in growth if used in replacing fishmeal. In both kind of substitutions, the n-3/n-6 ratio in tissues was improving. 

The paper worthwhile to be considered for the publication in “Animals” because the topic discussed is actual and interesting, mainly under a practical point of view.

Nevertheless, the paper should be revised making it more fluent and consistent, allowing the reader to follow the focus more easily.

Detailed points have been marked directly throughout the text.

Comments on the Quality of English Language

In the same time, the paper should be revised by an English native speaker to correct some formal mistakes.

Round 2

Reviewer 1 Report

Comments and Suggestions for Authors

Comments 4: L87 Increased or decreased EPA and DHA in muscle? The author of the above text explains that TM is deficient in EPA and DHA?

Response 4: The insect Tenebrio molitor (TM) is deficient in EPA and DHA, however, diets with animal and vegetable ingredients substitution by TM maintained and improved, respectively, EPA and DHA levels in muscle.

However, the introduction states that “Basto et al., [26] found that the complete substitution of FM by TM did not affect fish growth…” It is fishmeal not animal or vegetable ingredients substitution by TM. Please explain it.

Reviewer 2 Report

Comments and Suggestions for Authors

Thank you for considering the points of view discussed. I just suggest to include the idea that marine fish species are capable to retain n-3 fatty acids, due to their low ability to synthetize LC-PUFA.
